# Multifaceted confidence in exploratory choice

**Oleg Solopchuk** [1,2¤]*, **Peter Dayan** [1,2]

**1** Department of Computational Neuroscience, Max Planck Institute for Biological Cybernetics, Tübingen, Germany, **2** Department of Computer Science, University of Tübingen, Tübingen, Germany

¤ Current address: Institute of Systems Neuroscience, University Medical Center Hamburg-Eppendorf, Hamburg, Germany

* oleg.solopchuk@tuebingen.mpg.de

**Data Availability Statement:** Data and analysis code are available at https://github.com/solopchuk/confidence_exploration.

**Funding:** This work was funded by the Max Planck Society and the Alexander von Humboldt Foundation. PD is a member of the Machine

## Abstract

Our choices are typically accompanied by a feeling of confidence—an internal estimate that they are correct. Correctness, however, depends on our goals. For example, exploration-exploitation problems entail a tension between short- and long-term goals: finding out about the value of one option could mean foregoing another option that is apparently more rewarding. Here, we hypothesised that after making an exploratory choice that involves sacrificing an immediate gain, subjects will be confident that they chose a better option for long-term rewards, but not confident that it was a better option for immediate reward. We asked 250 subjects across 2 experiments to perform a varying-horizon two-arm bandits task, in which we asked them to rate their confidence that their choice would lead to more immediate, or more total reward. Confirming previous studies, we found a significant increase in exploration with increasing trial horizon, but, contrary to our predictions, we found no difference between confidence in immediate or total reward. This dissociation is further evidence for a separation in the mechanisms involved in choices and confidence judgements.

## Introduction

Making a choice is typically accompanied by a sense of confidence. Previous research has shown that we can express confidence about our perceptual- [1] or value-based choices [2], as well as specific choice attributes such as its value being a certain size (or a certain probability) [3] or the probability of a world state transitioning to a specific other world state [4]. However, there has yet to be any examination of attributional confidence judgments in a case in which a single choice engenders structural conflict. This arises in the exploration-exploitation dilemma in which optimal choices trade-off exploiting existing knowledge to pursue immediate rewards, and exploring to get information for the sake of future gains [5]. Longer decision horizons (i.e. more future choices) tip the balance towards exploration, while shorter horizons favour choosing the highest reward option known.

Human subjects behave in accordance with this dependency, with longer horizons leading to more uncertainty-directed, as well as more stochastic choices [6]. Furthermore, confidence judgments following exploratory choices are sensitive to estimates of option values and uncertainties about those values [3], which jointly determine their expectation about its

Learning Cluster of Excellence, EXC number 2064/
1–Project number 39072764 and of the Else
Kröner Medical Scientist Kolleg "ClinbrAIn: Artificial
Intelligence for Clinical Brain Research". The
funders had no role in study design, data collection
and analysis, decision to publish, or preparation of
the manuscript.

**Competing interests:** The authors have declared
that no competing interests exist.

consequences. In the relevant experiment, and as is conventional, participants were asked to report their confidence in their choice being correct [7]. However, as noted above, correctness in such decisions depends on the context: an exploratory action may be correct in anticipation of potential long-term gains, but wrong if it is necessary to maximise rewards here and now.

Here, we hypothesised that people's confidence judgments might be dissociated based on the context in which one and the same choice has to be evaluated. In two behavioural experiments involving a total of 250 participants, we administered a task known to elicit directed exploration as a function of the decision horizon [6], whilst simultaneously asking for ratings of confidence about short- and long-term rewards. We predicted that exploring, in the sense of foregoing a more certain reward for information about a lesser-known option, would be associated with lower confidence in immediate rewards and higher confidence in total rewards. This would indicate that people can flexibly assess confidence about the correctness of their choices with respect to different objectives.

## Materials and methods

### Participants

We recruited UK-based participants through the online platform Prolific in the period between 19.01.23 to 31.10.23. We filtered the submissions for performance ($>65\%$ accuracy in the short horizon condition; immediate and total reward confidence SD$>5$), which left us with 150 participants in experiment 1 (86M, 63F, 1 not indicated, mean age = 39.12) and 100 participants in experiment 2 (66M, 33F, 1 not indicated, mean age = 41.64). Participants were paid at a rate of £10 per hour, inclusive of a fixed £1 bonus. Participants provided written informed consent in accordance with procedures approved by the Ethics Committee of the Medical Faculty and Medical Clinic at the Eberhard-Karls-University of Tübingen (approval number 734/2019BO1).

### Task

We augmented the horizon task [6] with confidence judgements (Fig 1). The task was coded in jsPsych and adapted from https://nivlab.github.io/jspsych-demos/. In both experiments, each subject performed 112 trials in total. On each trial, participants were presented with 2 bandits and had to make 4 forced choices, followed by either 1 or 6 free choices (the 'horizon' manipulation). The forced choices resulted in either 2 samples from each bandit, or 3 samples from one bandit and 1 sample from the other bandit (information manipulation). Following the first free choice in the long horizon condition, subjects had to rate their confidence that their choice would lead to more immediate reward (one axis) and more total reward (other axis, counterbalanced). They made this report by putting a point down in a 2d square. The exact questions we asked, which had been the subject of extensive piloting, were "How confident are you that the choice you just made leads to a higher immediate payoff (i.e. in the 5th turn)?" and "How confident are you that the choice you just made leads to a higher total payoff (i.e. sum across the next 6 turns)?" They then executed the remaining 5 choices, but without reporting their confidence. In the short horizon condition, subjects only rated their confidence in getting more immediate reward, using a 1d scale. Following all free choices, they were shown the total amount of points, and proceeded to the next trial. To encourage more exploratory choices, we sampled both bandits following each free choice and only counted the points towards the total sum if the bandit the subjects chose gave more points than the one they didn't. Importantly, we showed the actual outcome of the bandit at each turn, so that the participants could not infer the value of the other bandit based on this threshold rule.

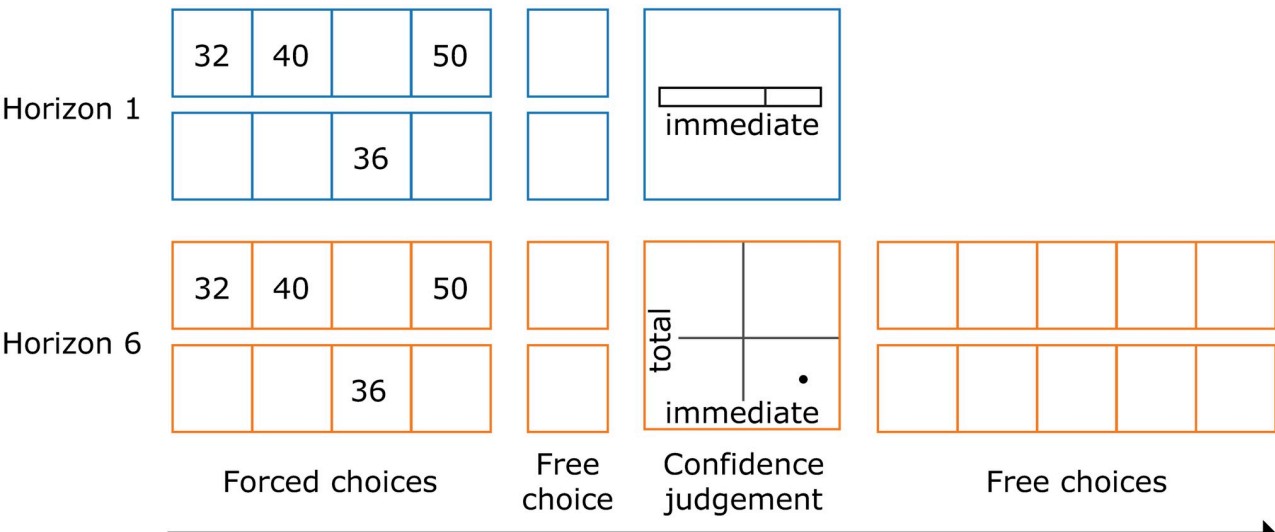

**Fig 1. Behavioural task.** Following four forced choices, participants could make either 1 (Horizon 1) or 6 free choices (Horizon 6). In the unequal information condition (presented here), 3 samples were shown for one bandit and one sample for the other. In the equal information condition, 2 forced choices were allocated to each bandit. A longer horizon encouraged participants to make more exploratory choices. Following the first free choice, the subjects had to rate their confidence in either getting more immediate and more total reward (on a 2d scale) or just getting more immediate reward (on a 1d scale).

The only difference between the two experiments was the range of the confidence scale. On one hand, we expect the strongest dissociation between confidence reports in short- and long-term rewards to occur in the part of the scale between 0 and 50, as this is when a short-term reward is sacrificed for exploration. On the other hand, previous studies have shown that the scale between 0 and 50 is less sensitive or reliable [8]. To account for both possibilities, we performed the experiment twice, with a scale from 0 to 100 in Experiment 1, and with a scale from 50 to 100 in Experiment 2. In Experiment 1, we instructed the participants that 0 corresponds to 'confident mistake', 50 to 'unsure', and 100 to 'certain'. In Experiment 2, we instructed the participants that 50 is 'guessing' and 100 is 'certain'.

## Analysis

We analysed subjects' choice behavior with the same computational model proposed by Wilson and colleagues [6]. This involves logistic regression, in which the choice between the two bandits, called A and B, is predicted by the difference between the mean values $\mu(A)$ and $\mu(B)$ of the samples available for the two, plus an information bonus $I$. Thus, the probability of choosing option A is given as

$$p(A) = \sigma(T^{-1}[\mu(A) - \mu(B) + \alpha * I(A)]), \tag{1}$$

where $\sigma(z) = 1/(1 + \exp(-z))$ is the standard logistic sigmoid, $T$ is the choice temperature, and $I(A)$ equals 1 if fewer previous samples were available for bandit A than bandit B (i.e. one sample), -1 if more were available, and 0 if both bandits had the same number of samples. An increase in information bonus parameter $\alpha$ with a longer decision horizon would indicate a directed exploration effect, while an increase in choice temperature would correspond to a random exploration effect (as choice would not specifically favour a less well-known option).

We analysed confidence judgements using both model-agnostic and model-based methods. For the former: focusing on long-horizon trials, we ran a 3-way ANOVA (uncertain chosen *

lower mean chosen * confidence question). Then, we narrowed the analysis down to trials in which participants chose an uncertain option that also had a lower average reward so far. This allowed us a maximal dissociation of exploratory from (immediate) reward-maximizing behaviour. We performed a Bayesian t-test on the difference of confidence about total reward and confidence in immediate reward in those trials, reporting Bayes factors [9]. Bayes factors are useful for quantifying the evidence in favour of both alternative and null hypotheses, as they represent the ratio of data likelihoods under these hypotheses. By convention, Bayes factors higher than 3/10/30 or lower than $^1/_3$/$^1/_{10}$/$^1/_{30}$/ are considered as some/strong/very strong evidence in favour of one of the hypotheses [9]. Here, we used $BF_{10}$, meaning that BF bigger than 3 or smaller than.3 indicated more evidence for the alternative or the null hypothesis, respectively.

In addition to the analysis above, we fitted the choice model to confidence judgements, minimising the squared error between the scaled probability of chosen option as predicted by the model and empirical confidence judgements (i.e., the backward fitting procedure proposed in [10]). We fitted the model separately to the judgments about immediate and total reward. A higher information bonus parameter for the model fit to the confidence about total reward would indicate that people were metacognitively sensitive to the fact that exploration may sacrifice short-term gains for later rewards. Finally, in order to account for the difference in confidence reporting between different subjects, we scaled the model probability with 2 extra parameters, following [10]:

$$\text{confidence(chosen)} = LB + (UB - LB) * p(\text{chosen}), \tag{2}$$

where *LB* and *UB* are lower and upper bounds respectively.

## Results

Consistent with earlier studies [6], we found a clear increase of exploratory behaviour in the long horizon condition (t-test on the difference between information parameters in long versus short horizon, Experiment 1: t(149) = 9.492248, p<.001, BF10 = 1.071e+14; Experiment 2: t(99) = 7.47, p<0.001, BF10 = 2.782e+08, t-test on the choice temperature difference quantifying random exploration, Experiment 1: t(149) = 7.77164, p = p<.001, BF10 = 6.225e+09, Experiment 2: t(99) = 6.22, p<0.001, BF10 = 9.401e+05; Figs 2 and 3). The 3-way ANOVA on confidence judgements revealed a significant effect of choosing a lower value option (Experiment 1: F = 181.36, p<0.001; Experiment 2: F = 126.61, p = p<0.001) and choosing an uncertain option (Experiment 1: F = 101.72, p<0.001; Experiment 2: F = 71.442, p<0.001). We found no significant difference between confidence about immediate and total rewards (Experiment 1: F = 0.0003, p = 0.99). Contrary to our prediction, we found no evidence for the 3-way interaction (chosen uncertain * chosen lower mean * confidence question) (Experiment 1: F = 0.004, p = 0.947; Experiment 2: F = 0.2273, p = 0.635).

The t-test on the difference of confidence that the choice was correct for total versus immediate rewards in trials in which subjects chose an uncertain option that also had a lower mean revealed no significant effect (Experiment 1: t(136) = 1.92, p = .056, BF10 = 0.57; Experiment 2: t(91) = 1.08, p = 0.282, BF10 = 0.203). Similarly, t-tests on the difference of information bonus parameters fit to the confidence about total and immediate rewards revealed no significant differences (Experiment 1: t(149) = -0.25, p = 0.8, BF10 = 0.094, Experiment 2: t(99) = 0.76, p = 0.444, BF10 = 0.147, Fig 3). We also found no difference in the temperature parameters when the model was fit to either total or immediate reward confidence (Experiment 1: t(149) = -0.50, p = 0.61, BF10 = 0.103, Experiment 2: t(99) = 0.72, p = 0.473, BF10 = 0.142, Fig 3). In addition, we tested the difference between the parameters fitted to choices (averaged

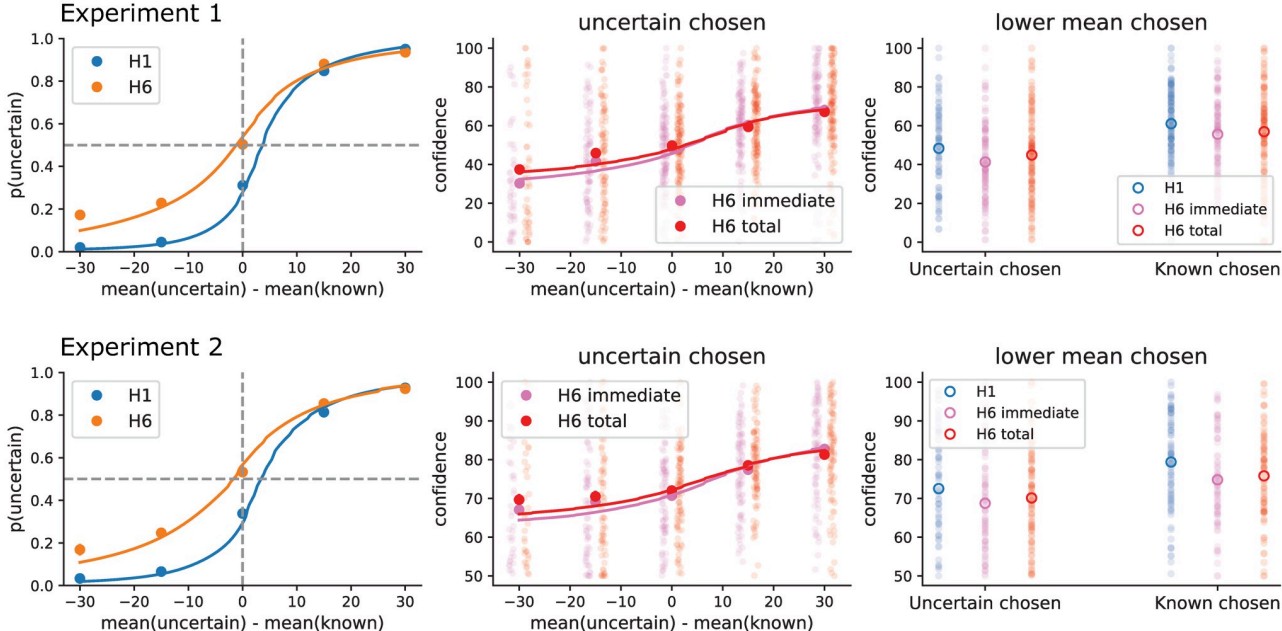

**Fig 2. Qualitative results.** The top panels correspond to experiment 1, where the subjects used a confidence scale ranging from 0 to 100, and the bottom panels correspond to experiment 2, in which a confidence scale between 50 to 100 was used. After finding a negative result in experiment 1, we reasoned that if the lower half of the scale is less sensitive, as suggested by some previous studies, we should offer people only the upper half of the scale. Both experiments agree in finding no evidence for context-dependence in confidence judgements following exploration. Left: the probability of choosing the uncertain option at the first free choice as a function of the difference between the means of uncertain and known option outcomes during the forced choices. The shift of the choice curve with increasing horizon is evidence for directed exploration. Middle: Confidence in getting more immediate or more total reward following the choice of the uncertain option, again as a function of mean difference between uncertain and known options. Right: Average confidence when the uncertain option was chosen and this option also had a lower empirical mean (average over the left half of the middle panel). Confidence at the short horizon is also shown for comparison. The curves of modeled choice and confidence were created by averaging subject-specific curves.

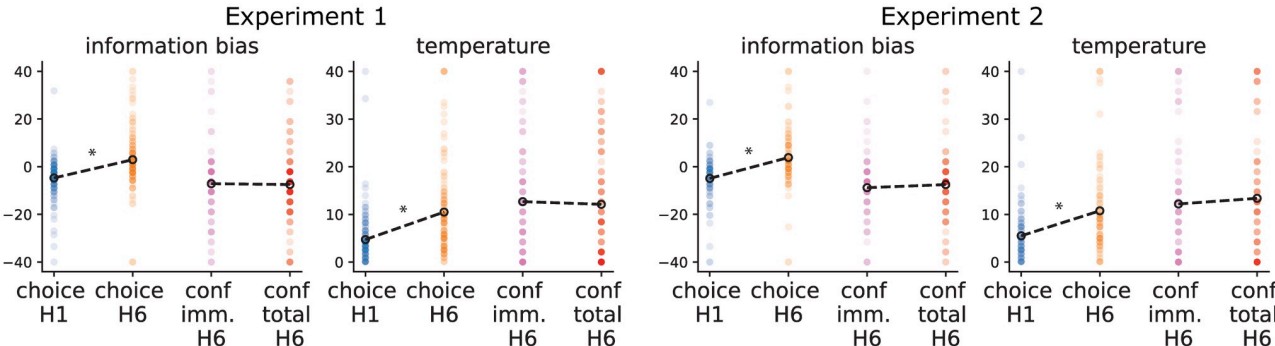

**Fig 3. Model parameter estimates.** Left panels: experiment 1, right panels: experiment 2. The figure shows the estimates of information bias and softmax temperature parameters fit to either choices in the short horizon, choices in the long horizon, confidence about immediate reward in the long horizon, and confidence about total reward in the long horizon. The model fit to choices also included the choice bias and the model fit to confidence had lower and upper bound parameters (not shown). The horizon-dependent increases in information bias and sigmoid temperature of the model fit to choices correspond to the directed and random exploration effect reported in the original study from Wilson and colleagues. The absence of difference in those parameters when fit to confidence in immediate vs confidence in total reward suggests the absence of the hypothesised difference in confidence with the context in which choices have to be evaluated.

between short and long horizons) and to confidence judgement (averaged between immediate and total reward questions in the long horizon). We found that the information-seeking parameter was significantly lower when fit to confidences as compared to when fit to choices (Experiment 1: $t(149) = 6.16$, $p<0.001$, BF10 = 1.507e+06; Experiment 2: $t(99) = 5.36$, $p<0.001$, BF10 = 2.418e+04), while the temperature parameter was significantly higher when fit to confidence as compared to when fit to choices (Experiment 1: $t(149) = -4.49$, $p<0.001$, BF = 1013.507; Experiment 2: $t(99) = -3.95$, $p<0.001$, BF10 = 128.093). This suggests that confidence ratings are less precisely tuned to choice accuracy than choices themselves, and is consistent with earlier studies [10].

We then tested whether instead of a group effect, there might be a between-subject correlation between any difference in confidence about immediate and total reward and choice behaviour. We did not find significant correlations of confidence difference with either the amount of exploration (difference in choice model information parameter between horizon, Experiment 1: $r(137) = 0.053$, $p = 0.53$, BF10 = 0.13; Experiment 2: $r(92) = 0.114$, $p = 0.278$, BF10 = 0.232) or with performance (average total number of points in the long horizon and unequal information condition, Experiment 1: $r(137) = 0.07$, $p = 0.40$, BF10 = 0.152; Experiment 2: $r(92) = 0.01$, $p = 0.912$. BF10 = 0.131). In addition, it could be that the difference in confidences about immediate and total reward gradually emerged over the course of the experiment as the subjects got more familiar with the task structure. Since some subjects made no or only a few purely exploratory choices (which we define as those with a lower mean and just one sample shown), we pulled such choices of all subjects together. We found no correlation between the difference in confidence about total and immediate reward and the trial number (Experiment 1: $r(547) = 0.05$, $p = 0.2$, BF10 = 0.121; Experiment 2: $r(385) = -0.06$, $p = 0.25$, BF0 = 0.122).

Finally, in the second experiment, we asked the subjects to report their strategies in making choices and confidence ratings. Surprisingly, we found out that only 6 subjects out of 100 reported anything related to information seeking or exploration in the long-horizon trials. This suggests that subjects might be implicit in their use of exploration, which in turn is consistent with their metacognitive unawareness of the long-term advantage of such choices.

## Discussion

Since the choices of human participants exhibit optimism in the face of uncertainty, and since choices are typically accompanied by somewhat calibrated judgments of confidence, we hypothesized that participants would be able to express dissociated confidence judgements—simultaneously having high confidence that a choice leads to more total reward and low confidence that the choice is better for more immediate reward. Although we confirmed earlier findings that confidence is sensitive to the difference of values of the chosen and unchosen options [2, 11], as well as to value uncertainty [3, 12, 13], contrary to our prediction we found that confidence judgements did not differ as a function of horizon.

There are several potential explanations for why we did not observe the expected confidence dissociation. One that we ruled out directly (via experiment 2) was the range of confidence scale. One might expect that on the trials on which the exploratory option has a lower expected immediate reward, post-exploration confidence dissociation would be found in the 0 to 50 part of the scale of the immediate judgements. On the other hand, this part of the scale has been previously shown to be noisy and unreliable [8]. Here, we indeed observed better calibration in experiment 2 (S1 Appendix), but did not find evidence for confidence dissociation in either variant—full scale in experiment 1 and half scale (50 to 100) in experiment 2.

Another potential reason for the lack of effect on average is that different individuals might use different information to shape their sense of confidence [14]. Similarly, people might differ in their metacognitive sensitivity, i.e. their sense of confidence might be tuned to performance to different extents [15]. For example, earlier studies have shown that people who are better metacognitively calibrated in tasks also perform them more proficiently [16]. To test for these possibilities, we assessed the correlation between the differences in average confidence about total and immediate reward during exploration and differences in choice behaviour such as average exploration and performance. The absence of any such correlation implies that an effect restricted to a subset of subjects is unlikely.

It could also be that instead of problems with the scale or metacognitive sensitivity, subjects had trouble understanding and dissociating between different questions. We tried to address this by piloting different ways to ask about confidence in immediate and total rewards, as well as by providing a real-life example in the instructions (see S1 Appendix).

Instead, we suggest that a perhaps more likely reason for the absence of the expected effect is that the subjects performed exploratory choices implicitly. In experiment 2, we asked the subjects to report their choice strategies following the completion of the experiment, and only a few subjects mentioned anything about information or uncertainty. This suggests one possible avenue for a future study in using a binary confidence scale, as it has been suggested that this can be more sensitive to implicit choices than continuous ones [17].

It could simply be that choices and confidence reports rely on different, even if overlapping, mechanisms [18], and that the information bonus calculation used to boost exploratory choices in the long horizon trials is not penetrable to metacognitive processing. This would be in line with previous studies reporting dissociable effects of task variables on choices and confidence judgements [11, 19].

## Conclusion

In sum, we found evidence against context sensitivity in confidence judgements concerning directed exploration. However, given the importance of metacognition in understanding normal and pathological decision making [20, 21], this remains an important direction for future research.

## Supporting information

**S1 Appendix.**
(DOCX)

## Acknowledgments

We are grateful to Andrew Webb and Sahiti Chebolu for helpful comments on the analysis code.

## Author Contributions

**Conceptualization:** Oleg Solopchuk, Peter Dayan.

**Data curation:** Oleg Solopchuk.

**Formal analysis:** Oleg Solopchuk, Peter Dayan.

**Funding acquisition:** Peter Dayan.

**Investigation:** Oleg Solopchuk, Peter Dayan.

**Methodology:** Oleg Solopchuk, Peter Dayan.

**Project administration:** Oleg Solopchuk.

**Resources:** Peter Dayan.

**Software:** Oleg Solopchuk.

**Supervision:** Peter Dayan.

**Validation:** Oleg Solopchuk.

**Visualization:** Oleg Solopchuk.

**Writing – original draft:** Oleg Solopchuk.

**Writing – review & editing:** Oleg Solopchuk, Peter Dayan.

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
