## [Decision Letter · Decision Letter 0]

12 Aug 2024

PONE-D-24-20376Multifaceted confidence in exploratory choicePLOS ONE

Dear Dr. Solopchuk,

Thank you for submitting your manuscript to PLOS ONE. After careful consideration, we feel that it has merit but does not fully meet PLOS ONE’s publication criteria as it currently stands. Therefore, we invite you to submit a revised version of the manuscript that addresses the points raised during the review process.

I received one review from a relevant expert who had a positive overall impression of the paper.  The reviewer lists a few queries that should be addressed in a revision.  I'll also note that it is stated in the manuscript that the data are publicly available, but I did not see a link in the paper.  Please include the link in your revision, along with a response to each comment made by the reviewer. 

We look forward to receiving your revised manuscript.

Kind regards,

Darrell A. Worthy, Ph.D

Academic Editor

PLOS ONE

Additional Editor Comments (if provided):

Reviewers' comments:

Reviewer's Responses to Questions

**Comments to the Author**

1. Is the manuscript technically sound, and do the data support the conclusions?

Reviewer #1: Yes

2. Has the statistical analysis been performed appropriately and rigorously? 

Reviewer #1: Yes

3. Have the authors made all data underlying the findings in their manuscript fully available?

Reviewer #1: No

4. Is the manuscript presented in an intelligible fashion and written in standard English?

Reviewer #1: Yes

5. Review Comments to the Author

Reviewer #1: Thank you for the opportunity to review this manuscript. In this study, the authors examine whether human participants will differ in their confidence for immediate versus long term gains depending on whether they opt to exploit or explore in a horizon task. Across two studies, the authors find no evidence for their prediction that exploration choices are associated with greater confidence for long-term rewards. This is a concise study with a clear hypothesis that is appropriately addressed by the data and analytical approach. I have some queries about the study that would improve the overall clarity and may help the authors consider other possible explanations of the data, which I outline below:

Introduction:

It would be helpful for a broader readership to briefly define key terms like decision horizon, as well as outlining the benefits of different explore/exploit strategies for short or long horizons.

Methods:

There is an absence of important demographic information about participants. For example, what was the age, gender distribution and ethnicity of participants? The age and gender would be particularly important, given age-related changes in exploration strategies, as well as evidence that males employ different explore/exploit strategies compared to females (Bach et al., 2021; Mata et al., 2013). Reporting the demographic information of participants would also be important to ensure the findings are reproducible.

Regarding the location of participants, were checks conducted to ensure IP addresses matched participants’ reported location?

Could the authors clarify how was the performance dependent bonus calculated?

Results:

Did the authors consider how the trial number was associated with self-report confidence for exploration choices? It may be that as participants become more familiar with the task structure they become better able to accurately estimate their confidence at increasing overall rewards over the task. Relatedly, it would be helpful in the methods to note how many trials the participants completed in total.

Another possible reason for the null findings is that the sample size was not large enough to detect a 3-way interaction. Did the authors conduct a power analysis to establish the sample size for this study?

I note the degrees of freedom for the correlations was 137 and 92 for Experiments 1 and 2, respectively (lines 150-153). Does that mean some participants did not have confidence data?

Discussion:

The discussion is clear and concisely summarises explanations for these findings.

6. PLOS authors have the option to publish the peer review history of their article (what does this mean?). If published, this will include your full peer review and any attached files.

Reviewer #1: No

---

## [Author Response · Author response to Decision Letter 0]

20 Sep 2024

Dear Editor, dear Reviewer,

Many thanks for considering our manuscript. We updated the submission with a link to publicly available code and data: https://github.com/solopchuk/confidence_exploration. Here are the point-by-point replies to the comments:

>Reviewer #1: Thank you for the opportunity to review this manuscript. In this study, the authors examine whether human participants will differ in their confidence for immediate versus long term gains depending on whether they opt to exploit or explore in a horizon task. Across two studies, the authors find no evidence for their prediction that exploration choices are associated with greater confidence for long-term rewards. This is a concise study with a clear hypothesis that is appropriately addressed by the data and analytical approach. I have some queries about the study that would improve the overall clarity and may help the authors consider other possible explanations of the data, which I outline below:

>Introduction:

>It would be helpful for a broader readership to briefly define key terms like decision horizon, as well as outlining the benefits of different explore/exploit strategies for short or long horizons.

We restructured the relevant introduction paragraph accordingly: 

‘Such conflict arises in the exploration-exploitation dilemma in which optimal choices trade-off exploiting existing knowledge to pursue immediate rewards, and exploring to get information for the sake of future gains (Gittins ‘79). Longer decision horizons (i.e. more future choices) tip the balance towards exploration, while shorter horizons favour choosing the highest reward option known. Human subjects behave in accordance with this dependency, with longer horizons leading to more uncertainty-directed, as well as more stochastic choices (Wilson et al ‘14). ‘

>Methods:

>There is an absence of important demographic information about participants. For example, what was the age, gender distribution and ethnicity of participants? The age and gender would be particularly important, given age-related changes in exploration strategies, as well as evidence that males employ different explore/exploit strategies compared to females (Bach et al., 2021; Mata et al., 2013). Reporting the demographic information of participants would also be important to ensure the findings are reproducible.

Thank you for pointing this out, we have updated the methods section with the demographics data: 

‘We filtered the submissions for performance ($>$ 65 $\\%$ accuracy in the short horizon condition; immediate and total reward confidence SD$>$5), which left us with $150$ participants in experiment 1 (86M, 63F, 1 not indicated, mean age = 39.12) and $100$ participants in experiment 2 (66M, 33F, 1 not indicated, mean age = 41.64).’

>Regarding the location of participants, were checks conducted to ensure IP addresses matched participants’ reported location?

We used the platform 'Prolific' that checks IP addresses according to this excerpt from their website: "IP addresses are checked, then identities verified with Onfido’s bank-grade ID checks.". 

>Could the authors clarify how was the performance dependent bonus calculated?

The participants were instructed that their payment would depend on both choices and confidence judgement as follows:

"Your payment for taking part in this experiment is composed of 2 equal halves: 1. A computer program will choose a game you played randomly, and the points you earned will be converted to money. You can increase your chances to get a high payoff for this half if you collect a lot of points in as many games as possible. 2. A computer program will choose a game randomly, and reward you based on the confidence judgement you made in that game, using a lottery. The payoff of the lottery is set up in such a way that you can increase your payoff chances by evaluating and reporting your real confidence as accurately as possible in as many games as possible."

Despite the instruction, we gave everyone a fixed bonus of 1 pound. We updated the methods to clarify this and included the instruction text above in the Appendix.

>Results:

>Did the authors consider how the trial number was associated with self-report confidence for exploration choices? It may be that as participants become more familiar with the task structure they become better able to accurately estimate their confidence at increasing overall rewards over the task. Relatedly, it would be helpful in the methods to note how many trials the participants completed in total.

Thank you for the suggestion. Since some participants made no or only a few purely exploratory choices (which we define as those with a lower mean and just one sample shown), we pulled such choices of all subjects together. We found no correlation between the difference in confidence about total and immediate reward and the trial number(Experiment 1: r(547) = 0.05, p = 0.2, BF10 = 0.121; Experiment 2: r(385) = -0.06, p = 0.25, BF0 = 0.122). We now include this analysis in the manuscript and additionally provide the corresponding scatter plots below: 

‘In addition, it could be that the difference in confidences about immediate and total reward gradually emerged over the course of the experiment as the subjects got more familiar with the task structure. Since some subjects made no or only a few purely exploratory choices (which we define as those with a lower mean and just one sample shown), we pulled such choices of all subjects together. We found no correlation between the difference in confidence about total and immediate reward and the trial number (Experiment 1: r(547) = 0.05, p = 0.2, BF10 = 0.121; Experiment 2: r(385) = -0.06, p = 0.25, BF0 = 0.122).’

[Figure in the response file]

We also updated the methods section with the number of trials (112 in both experiments): ‘In both experiments, each subject performed 112 trials in total.’

>Another possible reason for the null findings is that the sample size was not large enough to detect a 3-way interaction. Did the authors conduct a power analysis to establish the sample size for this study?

Following the completion of experiment 1, we estimated how many more subjects we would have needed to collect to get a decisive Bayes factor for the difference in confidence about immediate and total reward following an exploratory choice. For each of 10000 simulations, we appended the original dataset with N extra sampled subjects, calculated the Bayesian t-test, and recorded the resulting Bayes factor. We then decided to collect 100 subjects as this would result in a less than 10% chance of obtaining an inconclusive result (see the last panel of the figure below). We also followed a suggestion to restrict the confidence scale to 50 to 100 for a better calibration. These together gave rise to experiment 2, with its conclusive negative result.

[Figure in the response file]

>I note the degrees of freedom for the correlations was 137 and 92 for Experiments 1 and 2, respectively (lines 150-153). Does that mean some participants did not have confidence data?

The reason for reduced degrees of freedom is that some participants did not make any purely exploratory (lower mean and 1 sample shown) choices in which we expected the confidence judgments to be dissociated.

>Discussion:

>The discussion is clear and concisely summarises explanations for these findings.

---

## [Editor Report · Decision Letter 1]

3 Oct 2024

Multifaceted confidence in exploratory choice

PONE-D-24-20376R1

Dear Dr. Solopchuk,

We’re pleased to inform you that your manuscript has been judged scientifically suitable for publication and will be formally accepted for publication once it meets all outstanding technical requirements.

I think you have adequaltely addressed the concerns raised by the reviewer.

Kind regards,

Darrell A. Worthy, Ph.D

Academic Editor

PLOS ONE
---

## [Editor Report · Acceptance letter]

11 Nov 2024

PONE-D-24-20376R1 

PLOS ONE

Dear Dr. Solopchuk, 

I'm pleased to inform you that your manuscript has been deemed suitable for publication in PLOS ONE. Congratulations! Your manuscript is now being handed over to our production team.

Kind regards, 

on behalf of

Dr. Darrell A. Worthy 

Academic Editor

PLOS ONE